# A Protocol for Fabrication and on-Chip Cell Culture to Recreate PAH-Afflicted Pulmonary Artery on a Microfluidic Device

**DOI:** 10.3390/mi13091483

**Published:** 2022-09-07

**Authors:** Tanoy Sarkar, Trieu Nguyen, Sakib M. Moinuddin, Kurt R. Stenmark, Eva S. Nozik, Dipongkor Saha, Fakhrul Ahsan

**Affiliations:** 1Department of Pharmaceutical and Biomedical Sciences, California Northstate University College of Pharmacy, Elk Grove, CA 95757, USA; 2Department of Pediatrics and Medicine, Cardiovascular Pulmonary Research Laboratories, University of Colorado Denver, Anschutz Medical Campus, Aurora, CO 80045, USA; 3MedLuidics, Elk Grove, CA 95757, USA

**Keywords:** chip fabrication, PDMS-based chip, PAH-on-a-chip, PAH-chip, pulmonary arterial hypertension, pulmonary arterial cells on chip

## Abstract

Pulmonary arterial hypertension (PAH) is a rare pulmonary vascular disease that affects people of all ethnic origins and age groups including newborns. In PAH, pulmonary arteries and arterioles undergo a series of pathological changes including remodeling of the entire pulmonary vasculatures and extracellular matrices, mis-localized growth of pulmonary arterial cells, and development of glomeruloid-like lesions called plexiform lesions. Traditionally, various animal and cellular models have been used to understand PAH pathophysiology, investigate sex-disparity in PAH and monitor therapeutic efficacy of PAH medications. However, traditional models can only partially capture various pathological features of PAH, and they are not adaptable to combinatorial study design for deciphering intricately intertwined complex cellular processes implicated in PAH pathogenesis. While many microfluidic chip-based models are currently available for major diseases, no such disease-on-a-device model is available for PAH, an under investigated disease. In the absence of any chip-based models of PAH, we recently proposed a five-channel polydimethylsiloxane (PDMS)-based microfluidic device that can emulate major pathological features of PAH. However, our proposed model can make a bigger impact on the PAH field only when the larger scientific community engaged in PAH research can fabricate the device and develop the model in their laboratory settings. With this goal in mind, in this study, we have described the detailed methodologies for fabrication and development of the PAH chip model including a thorough explanation of scientific principles for various steps for chip fabrication, a detailed list of reagents, tools and equipment along with their source and catalogue numbers, description of laboratory setup, and cautionary notes. Finally, we explained the methodologies for on-chip cell seeding and application of this model for studying PAH pathophysiology. We believe investigators with little or no training in microfluidic chip fabrication can fabricate this eminently novel PAH-on-a-chip model. As such, this study will have a far-reaching impact on understanding PAH pathophysiology, unravelling the biological mystery associated with sexual dimorphism in PAH, and developing PAH therapy based on patient sex and age.

## 1. Introduction

Pulmonary arterial hypertension (PAH) is a progressive disorder characterized by vascular wall remodeling due to aberrant proliferation of endothelial and smooth muscle cells and chronic inflammation in the pulmonary arteries of lungs [1]. Animal models do not truly represent the pathophysiology of human PAH and show great variability in disease severity, histopathology, response to therapeutic interventions, and sex disparity [2,3,4,5,6]. Moreover, these models do not conform to the 3-R principle (replace, reduce, and refine) concerning the humane use of animals, although the FDA 2021 Modernization Act encourages use of non-animal models. In addition to animal welfare concerns, animal models are expensive, time consuming, and require complex invasive surgery [7,8]. Two-dimensional co-cultures and tissue-engineered 3D models cannot reproduce physical factors such as fluid stress and strain that impact organ functionality and thus are of limited use [9]. Overall, none of the existing animal or cellular models fully recapitulate the disease in humans.

**Figure 1 micromachines-13-01483-f001:**
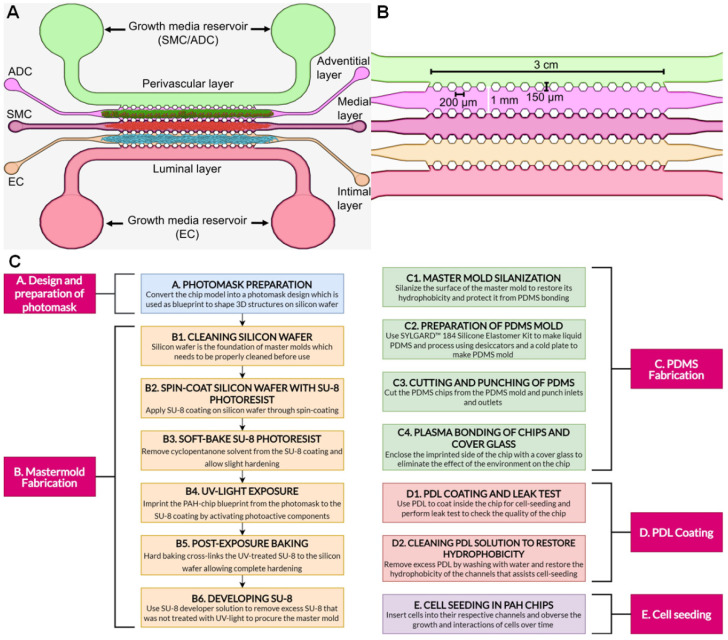
(**A**,**B**) PAH-chip design. (**A**) Five straight microchannels mimicking the biological structure of a pulmonary artery: growth media for smooth muscle cells (SMCs) and adventitial cells (ADCs) in the perivascular layer, growth media for endothelial cells (ECs) in the luminal layer, ADCs in the adventitial layer, SMCs in the medial layer, and ECs in the intimal layer. (**B**) Dimensions of the PAH-chip: The distance covered by pillars within the chip is 3 cm, inter pillar distance is 200 µm, the pillar height is 150 µm, and each set of pillars is separated by 1 mm. (**C**) A step-by-step flowchart of the PAH-chip manufacturing process.

Recently innovated state-of-the-art microfluidic technology can create miniaturized experimental systems referred to as “organ- or disease-on-a-chip” devices that allow recapitulating the functional units of various organs and disease conditions [10]. Recently, our group developed a multilayer microfluidic device, called PAH-on-a-chip (PAH-chip hereafter), that supports the growth of three major cells of PAH-afflicted pulmonary artery/arteriole: endothelial (PAH-EC), smooth muscle (PAH-SMC), and adventitial (PAH-ADC) cells [11]. This device emulates the major clinical features of PAH, such as arterial remodeling, muscularization, plexiform lesions, and sex disparity [11,12]. The PAH-chip structure consists of five microchannels mimicking five layers of a pulmonary artery: luminal, adventitial, smooth muscle, endothelial, and perivascular layers (Figure 1A). Structurally, the five straight channels are separated by hexagonal pillars with cavities (Figure 1B) to functionally copy the movement of cells from one channel to another. Here, we described a step-by-step method (Figure 1C) of PAH-chip design and fabrication and cell seeding on the device so investigators without an engineering background can prepare the chips, grow PAH cells, study the PAH pathophysiology, and screen anti-PAH drugs. Although there are several published protocols on polydimethylsiloxane (PDMS)-based chips [13,14,15,16,17,18,19,20,21,22,23], no such device is available that recapitulates the PAH-afflicted pulmonary artery. Importantly, protocol elaborated in this paper addresses the methodology for a new PDMS chip with a new application. Unlike many published studies, here we explain very small details and put cautionary notes, so investigators can accurately reproduce our PAH-chip device. In fact, this is the first multichannel chip that can emulate three major cell layers of a pulmonary artery in addition to luminal and perivascular layers. Currently, there are no protocols that detail the methodology for PAH-chip fabrication and on-chip cell seeding. This miniaturized device will allow researchers to better understand the processes implicated in the disease, expedite drug discovery, and help develop personalized therapy based on an individual’s sex and age. As such, we believe the methodology described in this study will spur a growth on the use of the microfluidic model for PAH, an under-investigated disease.

## 2. Safety Instructions

All manufacturing processes must be done inside a fume hood to maintain sterility, low concentration of airborne particles, and avoid contact with harmful chemicals. This practice will significantly reduce the dust particles that can settle on the surface of the master mold or PDMS chips at any point of the preparations, which can effectively increase the sterility of the chip, leading to more reliable experiments that are free from contaminants. Although, it is preferable to have lab conditions certified with clean room technology to avoid these problems altogether. Due to the use of photoresists during master mold fabrication, it is important to ascertain lithographic room conditions, which is a room fitted with yellow light or covering white light with yellow transparent acrylic sheet since most photoresists are sensitive to white light with a higher frequency. Neoprene gloves, safety glasses, acid apron, masks, and hair protectors should always be worn at all times to continue the effort of minimized contamination. When working with silicon wafers, SU-8, glass/plastic Petri dishes, and dissolving solutions, it is essential to wear masks to prevent contaminants from traveling onto the working area or machines. It is also vital to make sure the work area is always wiped clean. Most of the solvents and chemicals used need to be stored in specified conditions. Common cautions to perform before using any machines are mentioned throughout the protocol, such as the chronological steps to be followed to operate the UV-KUB 2 and cleaning Laurell spin coater after every use, to name a few. Finally, double-check all information and steps to avoid repetition since deviation in preparation can lead to significant irreversible damage.

## 3. Materials

### 3.1. Reagents

Photoresist SU-8 2100 (Kayaku, Cat. No. Y111075 0500L1GL)

Photoresist SU-8 developer solution (Kayaku, Cat. No. Y020100 4000L1PE)

Acetone (VWR)

99% Isopropyl alcohol-IPA (VWR)

Polydimethylsiloxane, PDMS-Sylgard™ 184 Silicone Elastomer Kit (DOW Corning)

Chlorotrimethylsilane, 98% (Thermo Scientific, Cat. No. 10214520)

Deionized water (Barnstead Mega-Pure D2, Thermo Scientific)

Sterile 10× phosphate buffered saline (PBS), pH 7.4 (Fisher Scientific, Cat. No. 70011044)

Phenol red sodium salt (Sigma-Aldrich, Cat. No. 14080-055)

NaOH tablets

Poly-D-lysine (PDL) hydrobromide (Sigma-Aldrich, Cat. No. P7886)

SmGM™- 2 Smooth Muscle Cell Growth Medium -2 BulletKit™ (Lonza, Cat. No. CC-3182)

EGM™-2 MV Microvascular Endothelial Cell Growth Medium-2 BulletKit™ (Lonza, Cat. No. CC-3202)

Invitrogen™ CellTracker™ Blue CMHC Dye, CELLTRACKER™ BLUE CMHC by Invitrogen™ C2111 (Fisher Scientific, Cat. No. C2111)

Invitrogen™ Qtracker™ 605 Cell Labeling Kit QTRACKER 605 CELL LABELING KIT by Invitrogen™ Q25001MP (Fisher Scientific, Cat. No. Q25001MP)

Invitrogen™ Qtracker™ 525 Cell Labeling Kit by Invitrogen™ Q25041MP (Fisher Scientific, Cat. No. Q25041MP)

Collagen Type I, rat tail (Sigma-Aldrich, Cat. No. 08-115)

### 3.2. Materials & Supplies

4″ (100 mm) Diameter Silicon Wafer (University Wafer)

4″ Diameter Single Wafer Carrier Box (MTI Corporation, SP5-S4)

Aluminum Foil

Wafer Handling Tweezers for Handling 4″ Wafers (Excelta, Cat. No. 490L-SA-PI)

5″ Glass Petri Dish (Cole Parmer, Cat. No. EW-34551-05)

Safety Wash Bottles, Low-Density Polyethylene, Wide Mouth (VWR)

Whatman Grade 2 Qualitative Filter Paper (TISCH Scientific, Cat. No. 1002-090)

P10 Micropipettes

P10 Barrier pipette tips (Santa-Cruz, Cat. No. sc-201721)

Tri-Cornered Polypropylene Beakers (Fisher Scientific, Cat. No. 14-955-111A)

Glass Stirring Rods (VWR, Cat. No. 59060-105)

Cutting Mat (Newark, PKN6003)

Scalpel Handle-Straight (Excelta, Part # 177-SE)

Scalpel Blades #10 Sterile for 181-SE Handle-Round (Excelta, Part # 181-10)

Plastic Petri Dishes with Clear Lid (Fisher Scientific, Cat. No. FB0875714)

Integra™ Miltex™ Standard 1 mm Biopsy Punches (Fisher Scientific, Cat. No. 12-460-401)

Integra™ Miltex™ Standard 4 mm Biopsy Punches (Fisher Scientific, Cat. No. 12-460-409)

Style 5 Ultra-fine pointed tweezers (Excelta, Cat. No. 5-SA)

Microscope Cover Glass: Rectangles (Fisher Scientific, Cat. No. 12-545-AP 30 × 22 mm)

Tissue Path Superfrost™ Plus Gold Slides (Fisher Scientific, Cat. No. 22-035813)

Cleanroom wipes (Fisher Scientific, Cat. No. 17-444-001)

Eppendorf tube 1.7 mL

Spatula (Aozita)

Parafilm (Parafilm M)

Luer Lock 10 mL Sterile Syringes (VWR, Cat. No. 53548-023)

Syringe Filter, PVDF, 0.22UM (Fisher Scientific, Cat. No. NC0992876)

### 3.3. Equipment

Spin Coater (Laurell Technologies Corporation, WS-650-23)

Wafer Alignment Tool (Laurell Technologies Corporation)

Nitrogen gas tank

Solid State Heat/Cool Cold Plate (Teca, Model AHP-1200CPV)

UV-LED Masking System: UV-KUB 2 (KLOÉ)

High Power Expanded Plasma Cleaner (HarrickPlasma, Cat. No. PDC-001-HP (115V))

UV Light Box Benchtop Decontamination Chambers (Airscience)

Level (instrument)

Weighing Scale (Fisher Scientific)

Vortex Mixer (VWR)

2 Vacuum Desiccators (Fisher Scientific)

Autoclave (Market Forge Industries, Model STM-EL)

Isotemp™ 500 Series Economy Lab Ovens (Fisher Scientific, Cat. No. 13246516GAQ)

Incubator at 37°, 5% CO_2_ (VWR)

Water bath at 37° (Precision)

Disposable Glass Pipets, 9 Inch (Fisher Scientific, Cat. No. 50-136-7739)

Biological Microscope XSZ-PWN107 (NingBo Proway Optics & Electronics Co., Ltd.)

LEICA DMi8 Microscope (Leica Microsystems)

### 3.4. Reagent Compositions

PDMS Liquid MoldWeigh 40 g of the SYLGARD™ 184 elastomer base and 4 g of the SYLGARD™ 184 curing agent on a disposable plastic beaker at 10:1 weight ratio. **NOTE:** Different amounts of elastomer base can be used while maintaining the 10:1 ratio. The significance of using 40 g of the elastomer base and 4 g of the elastomer curing agent is to cover the entire surface area of the master mold and create a thickness of 5 mm PDMS mold. Use a cleaned glass rod (rinse with acetone and air dry it with compressed air) to thoroughly mix the two elastomers in a circular motion for 3 min, resulting in a completely white and opaque mixture.PDL SolutionWeigh PDL powder at a ratio of 10 mg PDL per 20 microfluidic chips in a sterile 1.7 mL microcentrifuge tube. Add 1 mL sterile distilled water per 10 mg PDL to prepare 10 mg/mL solution and mix using a vortex mixer for 30 s. Filter sterilize the PDL solution with a 0.22 µm filter to remove any unwanted or insoluble particles. **NOTE:** If the insoluble particles are not removed, they may clog the chip microchannels, preventing liquid flow and incur leaks.Phenol red with PBS Dissolve 79.5 mg of Phenol red sodium salt in 500 mL of 10× phosphate buffered saline (PBS) using a vortex generator. This is required to make the collagen solution. 0.5 M Sodium hydroxide Dissolve 5 g of NaOH tablets in 230 mL of sterile deionized water by stirring. **CAUTION**: The solution will become hot if more than 1 g of NaOH tablet is added, therefore, add less than 1 g to deionized water at a time. This is required to make the collagen solution.Collagen solution, pH 7.4Mix 20 µL of 10× PBS, 105 µL of type 1 collagen, 52.5 µL of deionized water, 20 µL of cell suspension to the solution, and 2.5 µL of NaOH. **CAUTION:** Keep all components in ice to avoid solidification of the collagen solution. Mix slowly but thoroughly and avoid creating bubbles. 

## 4. PROCEDURE

### 4.1. Preparation of Photomask

A microfluidic chip consists of a set of microchannels etched or molded into a material such as glass, silicon, or polymer such as PDMS. The microchannel network design of the PAH-chip must be precisely elaborated based on biological functions and structures of the pulmonary artery and converting that into a design in 2D-modeling software such as Solidworks or Autodesk. Once the design is finalized, it is transferred to a photomask where the microscopic pattern of the chip is etched, leaving some regions transparent and some regions opaque. Photomasks are used as the blueprint for the chip structures to be mirrored onto (Figure 2A) and can be done with dedicated manufacturers. Photomasks are commonly produced in either chrome coated glass plates or plastic films, but the final choice depends on the budget and application. The steps below explain how to manually process a plastic photomask, although a chrome plated photomask (which does not need any manual processing) was used in this protocol.

A1Prototype a chip model based on biological structures and functions of the PAH-afflicted pulmonary artery/arteriole.A2Design the chip structures with a network of channels, pillars, inlets, and outlets on Solidworks (Dassault Systèmes).A3Convert the 3D model into a 2D stereolithographic (STL) file.A4Send the STL file to CAD/Art Services, Inc. to manufacture the photomask.A5Upon receiving the photomask, cut it into a square shape to match the dimensions of the thick cover glass provided inside the UV-KUB machine. **NOTE:** The thick cover glasses can be ordered directly from KLOÉ.A6Place and attach the photomask on the thick cover glass using one piece of scotch tape on each side (Figure 2B). **NOTE:** Do not block any lines, transparent or opaque, on the photomask while attaching tape.A7Put the photomask attached to thick cover glass in an airtight container to keep them away from incurring damage or accumulating dust particles. If the chrome-plated photomask is used (Figure 2C), skip steps A5 and A6.

### 4.2. Master Mold Fabrication

The master mold is fabricated using soft lithography following cautious and methodical steps (Figure 2D). The master mold essentially produces a mold that is used to fabricate a microfluidic chip capable of replicating various functions and mechanisms from different parts of the body. In this case, the basic materials needed to create a master mold are silicon wafer, 2100 grade SU-8 negative photoresist fluid, photomask with chip design, and chlorotrimethylsilane for the final coating. Some expensive equipment needed throughout the process of making a master mold is a silicon wafer alignment tool manufactured by Laurel, WS-650-23B Spin Coater, cold plates, and an ultra-violet ray treatment machine called UV-KUB 2. Below, we have presented in detail the process of master mold fabrication in six steps, B1 to B6:

#### 4.2.1. Cleaning the Silicon Wafer ● TIMING 25 Mins

The overall goal of this step is to remove specks of dust and smudges from the polished surface of the silicon wafer. Silicon wafers are thin circular slices of semiconductors that serve as the foundation of master mold fabrication. They are also called substrates, the underlying layer, for microfluidic devices manufactured in different diameters and thicknesses, as shown in Table 1. Overall, the wafer consists of two flat surfaces, a polished surface (Figure 2E) and an abrasive surface (Figure 2F), and two edges, a curved edge and a straight edge (Figure 2G). For PAH master mold preparation, 4-inch (100 mm) diameter silicon wafers are used with a thickness of 525 ± 10 µm (Figure 2G). In this protocol or any master mold fabrication, the polished surface is the area of interest that needs to be cleaned because of its adhesive properties to photoresist chemicals such as SU-8 negative photoresist.

B1.1Using wafer handling tweezers, grip a 4-inch diameter silicon wafer with the polished side facing up and place it on a glass petri dish. **NOTE:** (1) Whenever handling silicon wafers, grip the wafer with tweezers on the straight edge only (Figure 2G). Avoid using tweezers on the curved edge of the silicon wafer at any point of the fabrication process. (2) Discard any silicon wafers that has scratches, micro-cracks, or defects on the glossy surface.B1.2Pour ~20 mL acetone till the silicon wafer is fully submerged and soak the wafer for 20 min.B1.3Remove the wafer from the acetone solution using the tweezers and pour 10 mL 99% isopropanol to completely remove acetone from the glossy surface of the silicon wafer.B1.4Air dry both silicon wafer surfaces with nitrogen gas or compressed air.

**CAUTION:** Ensure not to touch the polished surface of the silicon wafer with or without gloves before or after cleaning the wafer. Proceed to step B2.1.

**PAUSE POINT:** The cleaned silicon wafers can be stored in wafer carrier boxes (air-tight containers) perpetually until use.

**Figure 3 micromachines-13-01483-f003:**
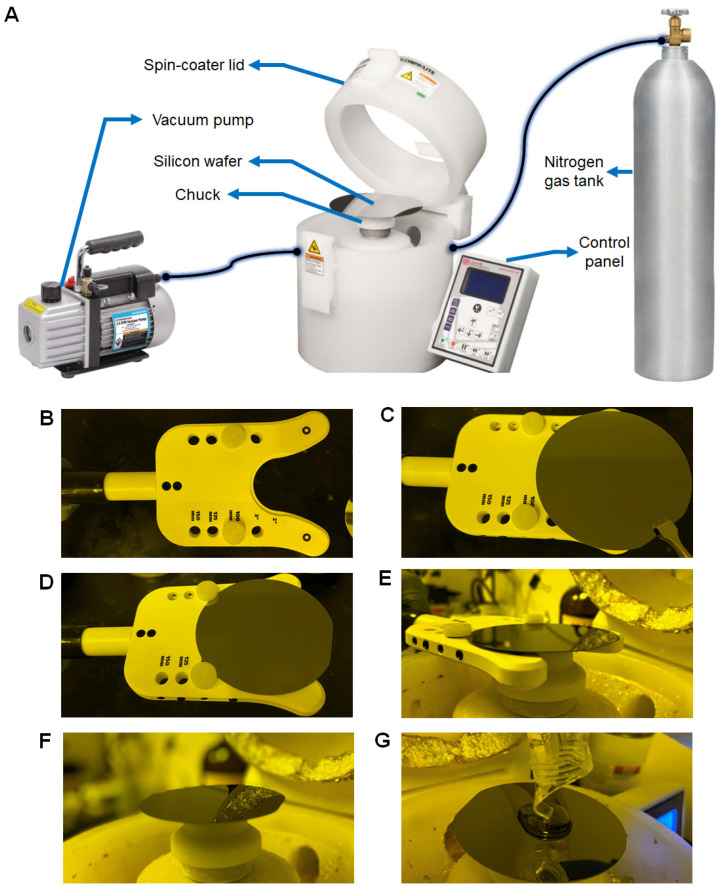
(**A**) Spin coater assembly—The spin coater is attached to a vacuum pump and a nitrogen gas tank using a proper tube and tube fittings. In addition, the control panel that comes with the spin coater is used to program the spin conditions, which helps evenly spread the SU-8 on the silicon wafer surface. (**B**–**G**) Handling and spin coating of the silicon wafer. (**B**) The plastic caps are set on 100 mm holes of the wafer alignment tool. (**C**) The silicon wafer is being put on the wafer holder with forceps. (**D**) The silicon wafer is snug fit into the plastic caps of the wafer holder. (**E**) The wafer holder is fit into the chuck and slowly lowered, which allows the silicon wafer to be positioned in the center of the chuck. (**F**) The abrasive surface of the silicon wafer is snug fit to the surface of the chuck due to vacuum. (**G**) Pouring SU-8 2100 on the center of the silicon wafer in a spiral formation avoiding the accumulation of bubbles.

#### 4.2.2. Spin-Coat the Silicon Wafer with SU-8 Negative Photoresist ● TIMING 10 Mins

This step aims to create a uniform thin film of SU-8, an epoxy-based, highly viscous gel-like negative photoresist, with 150 µm height on the cleaned, polished surface of the silicon wafer via a spin coater (Figure 3A). The significance of 150 µm thickness of SU-8 is to eventually provide a minimum height for channels in PAH-chip. There are several grades of SU-8 negative photoresist available, but specifically, two of them, SU-8 100 and SU-8 2100, can provide the desired thickness of 150 µm (Table 2). In this protocol, we used SU-8 2100 grade negative photoresist. Unlike positive photoresists, when SU-8 2100 negative photoresist is exposed to UV light, the chemical structure of the photoresist polymerizes. Regardless of the type of photoresist (positive or negative), the transparent region is the only region treated with UV light, meanwhile, the opaque region remains untreated since UV light cannot pass through that region. Therefore, the master mold’s channels are structured with SU-8 negative photoresist and the pillars are cavity regions, which is the inverse of the actual chip structure.

B2.1Set the plastic caps in the 100 mm diameter holes of the wafer alignment tool (Figure 3B).B2.2Grip the cleaned silicon wafer with tweezers (Figure 3C) and insert it in the plastic caps of the wafer alignment tool (Figure 3D).B2.3Turn on the vacuum and the nitrogen gas tank that is connected to the spin coater (Figure 3A).B2.4Open the lid of the spin coater (Figure 3A), position the wafer alignment tool towards the chuck of the spin coater and place the silicon wafer on the chuck (Figure 3E), and then remove the wafer alignment tool (Figure 3F). The vacuum will pull the silicon wafer towards the chuck. **NOTE:** If the silicon wafer is not placed correctly on the center of the chuck, the SU-8 spread will not be homogenous during spin-coating.B2.5Dispense 10 mL of SU-8 2100 photoresist on the center of the silicon wafer in a spiral formation (Figure 3G) to avoid the appearance of bubbles between the surface of the silicon wafer and the poured SU-8. **NOTE:** The 150 µm uniform height of SU-8 on the silicon wafer may not be achieved if higher or lower than the recommended amount of SU-8 is dispensed, leading to improper height differential.B2.6Close the lid of the spin coater. Set the program, as shown in Table 3, on the control panel of the spin coater and start spinning to achieve a uniform SU-8 2100 thin film with a thickness of 150 µm on the silicon wafer. **NOTE:** When the SU-8 is spun at high speeds, the centripetal force and the surface tension of the liquid together create an even covering on the silicon wafer. The two main factors that define film thickness are the spin speed and the viscosity of the solution. Other factors include spin time, solution density, solvent evaporation rate, and surface wettability.B2.7Once spinning is complete, turn off the vacuum, turn off the nitrogen tank, open the lid of the spin coater, remove the SU-8 2100 coated silicon wafer with tweezers, and proceed immediately to step B3.1. **NOTE:** Rotation continues until the excess SU-8 spins off the substrate and the desired thickness of the film is left on the substrate. Some excess SU-8 may remain on the edges of the silicon wafer which will not affect the master mold fabrication.

#### 4.2.3. Soft-Bake SU-8 Coated Silicon Wafer ● TIMING 1 H

After spin-coating, the thin film of SU-8 2100 negative photoresist on the silicon wafer remains gel-like and contains an organic solvent called cyclopentanone, which needs to be evaporated before imprinting the chip design. Thus, in this step, the SU-8 2100 coated silicon wafer is subjected to heat treatment, called soft-baking or first photoresist baking, for solvent evaporation, resulting in semi-solidification of the SU-8 2100 negative photoresist film.

B3.1Using a clean towel, clean the controllable heat/cool cold plate before use to minimize dust particles, because the presence of dust particles may change the temperature homogeneity of the SU-8 2100 coated silicon wafer.B3.2Lift the lid of the controllable heat/cool cold plate (Figure 4A). **!CAUTION:** The SU-8 thin film is in a liquid state, and thus, avoid tilting of the wafer during repositioning from the spin coater to the cold plate. **NOTE:** Ensure the cold plate is leveled.B3.3Place the SU-8 2100 coated silicon wafer on the controllable cold plate, ensuring that all parts of the wafer are sufficiently in contact with the cold plate (Figure 4A). **NOTE:** Ensure the entire abrasive surface of the silicon wafer contacts the controllable hot plate to preserve the uniformity of temperature. At any given point of soft baking, there should be little to no difference in the temperature at the center of the wafer and any corners of the wafer.B3.4Lower the cold plate lid to enclose the heat treatment region.B3.5Using TECA software, set the program such as set point, ramp time, soak time, function, and number of repeats, as shown in Table 4, and start the soft-baking process. **NOTES:** (1) It is important to gradually increase and decrease the temperature. Sudden changes in temperature over a short period of time will lead to excessive tension within the thin SU-8 film procuring solid SU-8 film with breakages or cracks. (2) Functions and description of Teca software: ‘Step’ signifies every unique function that the machine has to carry out; ‘set point’ is the fixed temperature the machine has to maintain; ‘ramp time’ is the amount of time the machine should take to reach the setpoint temperature; ‘soak time’ is the amount of time during which ‘set point’ temperature is maintained; ‘function’ enables the machine to recognize what to do at the end of each ‘step’; and finally ‘# of repeats’ is the iterations of the ‘steps’.B3.6After the soft-baking process is complete, leave the wafer in the cold plate, and proceed immediately to step B4.1.

#### 4.2.4. UV Exposure of Soft-Baked SU-8 2100 Coated Silicon Wafer ● TIMING 10 Mins

The overall goal of this step is to imprint the PAH-chip design onto the surface of the soft-baked SU-8 2100 coated silicon wafer through UV exposure using a UV–KUB 2 machine, and this process is called UV-LED masking treatment. Different components of the UV–KUB 2 machine are presented in Figure 4B. The photomask has transparent and opaque regions (Figure 2A), and the UV light can only pass through the transparent regions, which is the imprint of the chip design. Once the UV light passes, it reacts with the soft-baked SU-8 2100 negative photoresist, resulting in the activation of the photoactive component (PAC) that initiates cross-linking (hardening) of the SU-8 2100 photoresist with the silicon wafer. The opaque region does not allow passage of UV light, thus there is no activation of PAC or cross-linking of SU-8 with the silicon wafer in that region.

B4.1Turn on the UV-KUB 2 machine. **!CAUTION:** (1) Turn on the machine only when using it and it is paramount to turn off the machine once masking treatment is done in order to avoid any internal permanent damage. (2) Do not place anything on top of the machine since this machine is specifically calibrated for masking treatment and any sort of force on top of the machine will alter the calibration, causing problems with the masking treatment. (3) The front screen monitor is very sensitive, therefore, only use the stylus provided by the manufacturer to select options from the front screen monitor.B4.2The front screen opens with three options: (i) masking, (ii) full surface, and (iii) settings. Press the ‘masking’ option.B4.3Pry the soft-baked SU-8 2100 coated silicon wafer from the hot plate using the tweezers from the straight edge side of the wafer (Figure 4C), and once wafer is detached from the hot plate, grip the SU-8 coated silicon wafer and place it on the loading tray (Figure 4B*i*) with the coated side in an upright position (Figure 4B*ii*).B4.4Place the photomask (Figure 4B*iii*) facing down on the support cylinders (Figure 4B*iv*) such that the photomask and the SU-8 2100 coated side of the silicon wafer are facing each other.B4.5Click the arrow on the screen of the UV-KUB 2 machine. This will prompt the loading tray to retract, and a new screen will appear called ‘adjust the distance for a 4-inch wafer’.B4.6Input ‘675’ for ‘Thickness’. This 675 μm thickness refers to the thickness of the silicon wafer (525 μm) plus thickness of the SU-8 2100 thin film (150 μm).B4.7Press ‘V’ to ‘Validate’. **NOTE:** Press ‘C’ on the right bottom corner of the screen to ‘Cancel’ and change the numbers if the input is wrong at any point.B4.8Input ‘4’ for ‘Masking Distance’, then press ‘V’ to ‘Validate’. **NOTE:** Masking distance of 4 μm refers to the distance between the photomask and the SU-8 2100 thin film. This masking distance ensures no contact between the two surfaces. Masking distance is also responsible for the resolution of the chip design imprint on the SU-8 2100 thin film. For instance, the higher the masking distance, the lower the resolution, and vice-versa.B4.9The machine is now auto-adjusting based on the values provided in steps B4.6 and 4.8. Once auto-adjustment is complete, the first line of the screen will show “Wafer in position”. Press ‘Continue’ to progress (or press ‘Cancel’ to go back and change numbers if need be).B4.10The new screen will now show two options: ‘New Cycle’ or ‘Memory’. Select ‘New Cycle’ when using the machine for the first time (or select ‘Memory’ and skip steps B4.11-B4.16 if the cycle has already been saved in the memory of the machine).B4.11This prompts two options: ‘Pulse’ or ‘Continuous’. Select ‘Pulse’.B4.12Input ‘2’ for ‘Cycles’. Input ‘5s’ for ‘Duration ON’. Input ‘1s’ for ‘Duration OFF’. Input ‘60’ for ‘Power (in %)’. **NOTES:** (1) One cycle consists of 5 s of UV treatment and 1 s of no treatment accounting for 6 s and this 6 s cycle is repeated once more time. (2) Maximum power can be used to lower the time, but the resolution of the structure will not be optimized. Therefore, the above listed power and cycle setting optimizes the sharp edges within the structure. The machine has a maximum capacity of 180 watts, meaning 180 Joules per second, which represents 100% power. Thus, 60% power of the machine is 108 Watts, which is 108 Joules of energy being applied to the surface of SU-8 per second. The amount of power needed may also vary by plus or minus 5 to get an exact whole number percentage. (3) Depending on the maximum angle in the structure, the cycle recommendations will change. In the case of trapezoidal structures, the angles that need to be reached are 45° and 135°; therefore, the cycle information will be based on optimizing the edges created from those two angles. To create angles of those values, the resolution needs to be optimized, which is done by changing the cycle information.B4.13Press the button on the right bottom corner of the screen to continue.B4.14A new screen appears that summarizes all inputs. Verify the information on the screen: ‘Cycles = 2’, ‘Duration ON = 05 s’, ‘Duration OFF = 01 s’, ‘Duration = 12 s’, and ‘Power (in %) = 60’. On the bottom of the screen, there are three options: ‘new’, ‘save’, and ‘insolate’. Press ‘save’ to save the program.B4.15The new screen shows cycles 1–10. Pick any cycle to save the program.B4.16After saving, press ‘Insolate’ which starts the UV-masking process (skip steps B4.17 and B4.18 if new cycle is selected in step 10).B4.17Now select the memory that has the cycle information stored.B4.18Press ‘Insolate’ to start the process of UV light treatment that imprints the chip design on the surface of the SU-8 2100-coated silicon wafer. After completion of UV-masking, press the bottom right button to open the loading tray. **NOTE:** Chip design imprint remains invisible on the surface of the UV-treated SU-8 2100. It becomes visible in step B5.B4.19Gently lift the photomask from any corner and store it in a clean closed container. **NOTE:** If the photomask comes into contact with the surface of the SU-8 2100 during UV-masking treatment, discard this silicon wafer.B4.20Using tweezers, grip the imprinted SU-8 2100 coated silicon wafer and immediately proceed to step B5.1.B4.21Select ‘Cancel’ on the screen of the UV-KUB 2 machine to retract the loading tray. This will also prompt the machine logo, indicating all the processes inside the machine are stopped. Turn off the UV-KUB 2 machine.

#### 4.2.5. Post-Exposure Baking (PEB) ● TIMING 2.5 H

In Step B4, the UV exposure initiates the cross-linking of SU-8 with the silicon wafer but needs further energy to complete the reaction. Thus, in this step, the silicon wafer is subjected to heat treatment through post exposure baking (PEB), also called hard baking, which facilitates complete cross-linking (hardening) of the UV-exposed SU-8 region.

B5.1Lift the cover of the controllable hot plate and place the imprinted SU-8 2100 coated silicon wafer such that all parts of the wafer are sufficiently in contact with the controllable hot plate (Figure 4A).B5.2Place the cover back on the controllable hot plate so that the evaporated SU-8, which is toxic, is contained inside and also to avoid the accumulation of dust particles on the surface of the SU-8 2100. **NOTE:** Any dust particles that settle on the UV-treated region of the SU-8 2100 may get permanently attached to it, adversely impacting chip fabrication.B5.3Turn on the controllable hot plate and physically connect the controllable hot plate to the computer containing the Teca software.B5.4Open the Teca software and insert the post-exposure baking values such as set point, ramp time, soak time, function, and number of repeats, as shown in Table 5.B5.5Click on ‘Save to Controller’ to save the program. Then, start the post-exposure baking. Once baking is complete, immediately proceed to step B6.1. **NOTE:** After 2 min of hard baking, the edges and corners of the chip start to become slightly visible. By the end of post-exposure baking, the imprinted structures of the chip, such as channel walls and pillar edges, become visible and can be seen with the naked eye.

#### 4.2.6. Washing the Hard-Baked SU-8 2100 Silicon Wafer with SU-8 Developer Solution ● TIMING 15 Min

The SU-8 region that is exposed to UV light is cross-linked (hardened) with the silicon wafer, and therefore, remains insoluble to the SU-8 developer solution, while the non-UV-exposed SU-8 region remains soluble. Here, the SU-8 developer solution is used to wash and remove the soluble SU-8 from the silicon wafer so that only the hardened area, i.e., chip imprint, remains.

B6.1Wash a 5″ glass petri dish with acetone in a fume hood to remove any contaminants, such as dust particles. Place the clean petri dish on a flat surface inside the fume hood and pour 15 mL of the SU-8 developer solution on the glass petri dish.B6.2Using tweezers, transfer the hard-baked SU-8 2100 silicon wafer from the controllable hot plate to the glass petri dish containing the SU-8 developer solution, ensuring that the top surface of the hard-baked SU-8 2100 is fully submerged in the developer solution.B6.3Leave the silicon wafer submerged in the developer solution for a total of 7–8 min. After the first 5 min of submergence, gently tilt the glass petri dish left and right for at least 10 iterations to facilitate removal of the soluble SU-8. **NOTE:** After 7–8 min, the 3D structure of the chip starts to become more visible.B6.4Using forceps, grab the silicon wafer and keep it slanted (Figure 4D) on a new glass petri dish and apply the SU-8 developer solution using an LDPE bottle to the SU-8 coated side of the silicon wafer to ensure removal of any remaining soluble SU-8. **NOTE:** Keep the glass petri dish containing the developer solution aside for further use in step B6.7.B6.5Spray the bottom corner of the slanted silicon wafer with 99% isopropyl alcohol (IPA) using an LDPE bottle to confirm if the soluble SU-8 is completely solubilized. Avoid hitting the chip structures with IPA in case the SU-8 is underdeveloped. If a white residue appears on the silicon wafer surface (Figure 4E), or if the IPA falling on the glass petri dish is emerging white residue (Figure 4F), the SU-8 is under-developed, therefore, immediately stop applying IPA and proceed to step B6.6. If no white residue emerges (Figure 4G), skip steps B6.6 and B6.7. **!CAUTION:** It is important not to over-wash or under-wash the master mold since the solution is applied everywhere and over-washing may lead to structural damage or under wash may lead to excess (soluble) SU-8 remaining on the structure.B6.6Rinse the silicon wafer with developer solution (Figure 4D) to flush out IPA to prevent contamination of the SU-8 developer solution (in step B6.7).B6.7Submerge the silicon wafer in the same glass petri dish containing the SU-8 developer solution (used in B6.2) for 1 min, and while submerged, tilt the glass petri dish gently left and right at least 10 times. Repeat step B6.4.B6.8Rinse the entire surface of the silicon wafer with IPA and then rinse with the SU-8 developer solution using an LDPE bottle.B6.9Repeat step B6.8 two more times to ensure complete clearance of the soluble SU-8.B6.10Gently apply compressed air to dry both surfaces of the silicon wafer, which completes the master mold fabrication process. **NOTE:** The source of compressed air can be nitrogen gas or compressor. **!CAUTION**: Be very cautious when spraying compressed air on the glossy surface of the silicon wafer as excessive air pressure may lead to detachment of the SU-8 2100 from the silicon wafer.B6.11Perform quality control of the master mold by observing it under a light microscope. The edges should be sharp under the microscope (Figure 4H), the channels should be linear (Figure 4I), and the inlet and outlets should be curved with no inconsistencies (Figure 4I). **NOTE:** There may be defects in the edges of the pillars and/or channels. If there are defects, some options to explore would be optimizing spin conditions, maintaining sterility of the silicon wafer, soft-baking and hard-baking parameters, alignment of the photomask with the SU-8 molded silicon wafer, and finally, the UV-KUB 2 inputs of thickness and masking distance.

**PAUSE POINT:** Place the master mold in a wafer carrier box or cleaned glass petri dish until use.

### 4.3. PDMS-Chip Fabrication

The master mold is an inverse structure of the chip. To fabricate the chip, PDMS is mixed with a curing agent and that mixture is poured onto the master mold. The mixture is then treated through various processes to solidify and eventually pried to create the chips (Figure 5A), as detailed below in four major steps:

#### 4.3.1. Silanization of Master Mold ● TIMING 1.5 H

Silanization is the covering of the master mold surface with a silane solution, Cholorotrimethysilane 98%, resulting in the formation of a hydrophobic silane monolayer which enhances the adherence of PDMS to the SU-8 and adds a layer of protection to allow easy removal of the PDMS mold, which is hydrophobic, from the surface of the master mold. Silanization encourages the overall surface of the master mold containing hydroxyl groups to be displaced with the alkoxy groups, thus forming a covalent -Si-O-Si- bond. The method of PDMS mold fabrication is discussed in detail in Section C2, which involves the addition of a mixture of two elastomers on the master mold surface to seep into the cavity regions of the master mold and form the chip structure. Silanization also ensures that the PDMS mold can be easily removed from the master mold without detaching the SU-8 from the silicon wafer.

C1.1Attach the master mold to the master mold glass petri dish using scotch tape (Figure 5B) to ensure that the master mold is static during silanization.C1.2Place the master mold inside desiccator I without the petri dish lid. **NOTE:** A desiccator is a glass container or other apparatus holding a drying agent for removing moisture from specimens and protecting them from water vapor in the air. It is also used to make sure that the corrosive chemical does not roam freely in the air as it can have a harmful effect on the eyes.C1.3Place a filter paper next to the master mold and put 100 µL of cholorotrimethysilane on the filter paper and close the desiccator with the desiccator lid (Figure 5H).C1.4Connect the desiccator to a vacuum pump. **NOTE:** The desiccator lid cannot be detached from the desiccator when a vacuum is created.C1.5Maintain the vacuum condition for 1 h to fully treat the surface of the master mold with a hydrophobic layer of cholorotrimethysilane. **CRITICAL STEP**: During this hour, proceed to Step C2 to simultaneously prepare a mixture of PDMS and curing agent.C1.6After 1 h, release the vacuum and cover the master mold with the petri dish lid to avoid any contamination with dust particles.

**Figure 5 micromachines-13-01483-f005:**
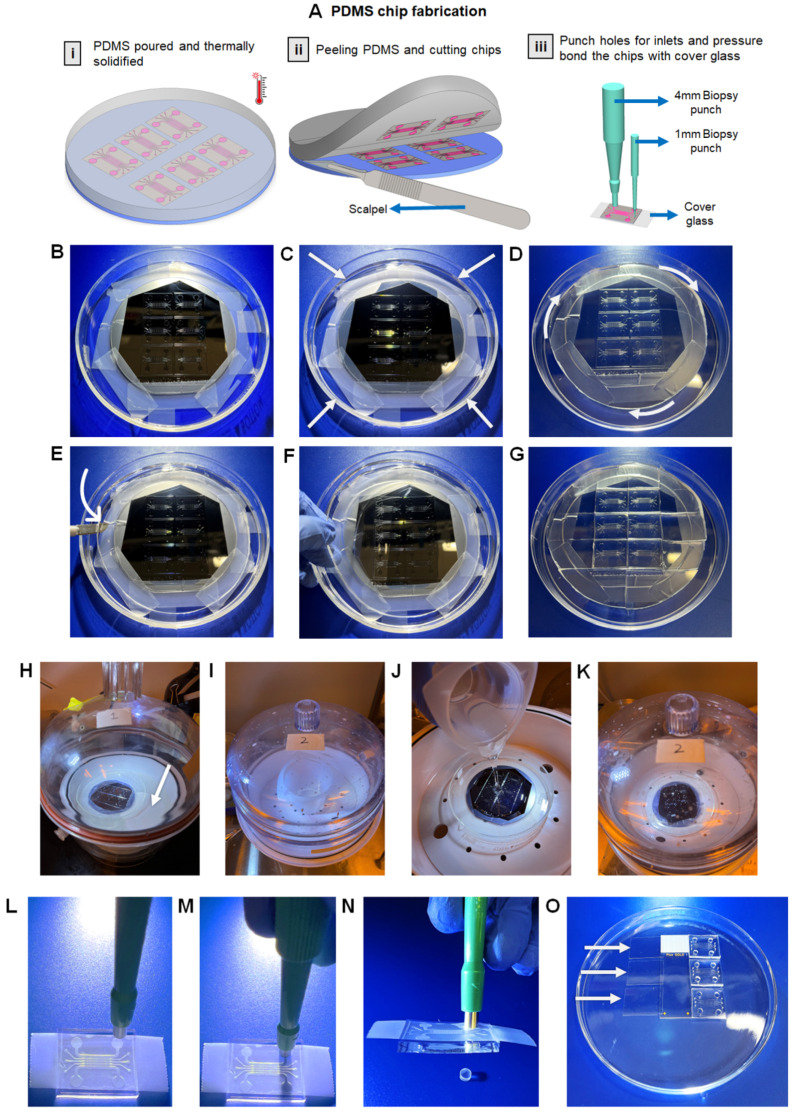
(**A**) A summary of PDMS chip fabrication—(**i**) Liquid PDMS is poured on the master mold, which is then thermally solidified on the cold plate, (**ii**) The PDMS mold is removed from the master mold and cut into the shape of the chips, (**iii**) Every chip is punched using biopsy punches and then plasma treated with cover glass to enclose the chip, which creates a closed environment for the PDL coating. (**B**–**G**) Preparation of PDMS mold—(**B**) The master mold is attached to a glass petri dish with pieces of tape, (**C**) Successful preparation of the PDMS mold on the master mold (layer of master mold is indicated by arrows), (**D**) Cutting the master mold around the edge (indicated by arrows) without damaging the silicon wafer to be able to remove the mold, (**E**) Using a scalpel to separate the PDMS mold slab from the glass petri dish, (**F**) Use gloved hand to pry the master mold slab and place it on a new plastic petri dish, (**G**) Make horizontal and vertical slices to cut chips out of the PDMS slab. (**H**–**K**) Desiccator configuration—(**H**) Desiccator I with master mold and filter paper (indicated by an arrow) for silanization, (**I**) Desiccator II with the plastic mug containing PDMS liquid. This desiccation will remove air bubbles from the PDMS liquid, (**J**) Pouring air bubble-free PDMS to the center of the silanized master mold in desiccator II, (**K**) Desiccation removes air bubbles from the poured PDMS. (**L**–**O**) Punching process of PAH-chips—(**L**) A piece of tape is used to be able to visibly recognize the location of the holes from the chip structures surface since PDMS is transparent, (**M**) Position and place the biopsy punch on top of the desired hole and use force to punch the hole on the chip, (**N**) With the puncher attached to the chip, lift the punch and press the button on top of the punch to remove excess PDMS and then remove the biopsy punch from the chip to create the hole on the chip. (**O**) Glass petri dish setup for plasma treatment—Place 3 pieces of sterile glass coverslips (indicated by white arrows) on one edge of the Superfrost glass slide and place 3 PAH-chips with the chip structures facing upwards on another edge of the Superfrost glass slide and ensure that there is no contact between the glass coverslips and the chips.

#### 4.3.2. Preparation of PDMS Mold ● TIMING 4 H

The goal of this step is to make the PDMS mixture from the SYLGARD™ 184 Silicone Elastomer Kit containing the SYLGARD™ 184 elastomer base and the SYLGARD™ 184 curing agent. This mixture is then poured on the master mold and heat treated to solidify and form chips. PDMS is a mineral-organic polymer (a structure containing carbon and silicon) of the siloxane family (a word derived from silicon, oxygen, and alkane), widely used for the fabrication and prototyping of microfluidic chips. The purpose of the elastomer base is to provide the bulk of the chip material and elasticity, whereas the elastomer curing agent facilitates the bonding of the molecular components of the elastomer base to the surface of the mixture through heat activation. During heat treatment, the two elastomers begin polymerization (also known as cross-linking), allowing the elastomer curing agent to be consumed by the elastomer base. The heat treatment also solidifies the liquid PDMS and helps build a 3D network of connections participating in the mixture’s chemical and mechanical properties. Without the curing agent, the particles inside the polymers would not be interconnected and remain unattached to the elastomer bases.

C2.1Weigh 40 g of the SYLGARD™ 184 elastomer base and 4 g of the SYLGARD™ 184 curing agent on a disposable plastic beaker at 10:1 weight ratio. **NOTE**: Different amounts of elastomer base can be used while maintaining the 10:1 ratio. The significance of using 40 g of the elastomer base and 4 g of the elastomer curing agent is to cover the entire surface area of the master mold and create a thickness of 5 mm for the PDMS mold. In addition, the 10:1 ratio is critical to maintain the elasticity of the PDMS mold. Generally, increasing the proportion of curing agent (e.g., a ratio of 8:1 for the elastomer base to the elastomer curing agent) enhances the cross-linking structure of the elastomers, leading to a reduction of the elasticity and increase in the rigidity of the PDMS mold, which can cause problems in cutting or punching the chips in steps C5 and C6. **!CAUTION:** Do not exceed 55 total grams of mixture in one plastic mug because overfilling may lead to forming of foam even when the desiccator is used, and the foam will keep recycling through the mixture, obliterating the purpose of removing air bubbles.C2.2Use a cleaned glass rod (rinse with acetone and air dry it with compressed air) to thoroughly mix the two elastomers in a circular motion for 3 min, resulting in a completely white and opaque mixture. **NOTE:** Hand mixing of elastomers will generate air bubbles that are removed in step C2.3.C2.3Place the plastic beaker containing the elastomer mixture in desiccator II (Figure 5I), connect it to a vacuum pump, and sustain a vacuum condition for at least 45 min. This process will remove air bubbles that were generated during step C2.2. Desiccation is complete once the mixture is free from air-bubbles and the desiccator lid looks transparent, which is when the pump is stopped, and the vacuum is released. **NOTE:** The amount of time the mixture should be placed in a vacuum depends on 3 factors: (i) the force of the pump, (ii) the amount of PDMS and curing agent mixture, and (iii) the amount of air trapped in the PDMS and curing agent due to mixing.C2.4Transfer the petri dish containing the master mold from desiccator I to desiccator II and pour the elastomer mixture on the center of the master mold. Allow steady and continuous flow of liquid PDMS to evenly spread the elastomer mixture to cover the entire surface of the master mold (Figure 5J). **NOTE:** It is important to pour the mixture very quickly to reduce formation of air bubbles between the master mold surface and liquid PDMS layer.C2.5Recreate vacuum condition for 1 h in desiccator II (Figure 5K) to remove any air bubble accumulation during pouring. Once desiccation is complete, release the vacuum and cover the PDMS-coated master mold with a petri dish lid.C2.6Transfer the petri dish into an oven and heat-treat for 2 h at 65 °C. **NOTE:** Ensure that the petri dish is leveled inside the oven to avoid tilted solidification of PDMS. If the petri dish is not level, the PDMS mixture, which is still in liquid state, will solidify with an uneven height.C2.7After 2 h, transfer the petri dish into a clean hood and cool down for 20 min at room temperature, which completes the solidification of the PDMS (PDMS mold hereafter) on the master mold (Figure 5C). **NOTE:** Heat treatment causes the silicon wafer to be brittle, therefore cooling is mandatory. When trying to remove the chip from the master mold, if not cooled properly, the master mold may break.

#### 4.3.3. Cutting and Punching of PDMS Mold ● TIMING 30 Min

The purpose of this step is to detach the solidified PDMS from the master mold, cut the PDMS mold into the shape of chips, and punch inlets and outlets on the PAH-chips. All the cutting is done on a glass petri dish or on a plastic petri dish placed on top of a cutting mat. The cutting mat delineates the boundaries of the chips which helps cut the mold into individual PAH-chips. Every PAH-chip contains 4 holes that are 4 mm in diameter and 6 holes that are 1 mm in diameter. The 4 mm holes are cut by 4 mm biopsy punch and the 1 mm holes are cut by 1 mm biopsy punches. These holes allow the flow of fluid through channels and provide enough space for micropipettes to introduce cell cultures.

C3.1Using a scalpel, cut the PDMS mold in a circular path around the outer edge of the silicon wafer, thus avoiding any damage to the master mold (Figure 5D). **NOTE**: One master mold can be utilized for the preparation of at least 50 PDMS molds if its surface remains unaltered.C3.2Pry the PDMS mold with the scalpel in the direction of the channels (Figure 5E), then grab and pull the PDMS mold with a gloved hand (Figure 5F) and place it on a new plastic petri dish with the PAH-chip structure facing upwards (Figure 5G). **CRITICAL STEP**: Removal of the PDMS mold in any direction other than towards the direction of the channels can rupture the channels and pillars of the chips. **NOTE:** Remove the excess PDMS mold from the glass petri dish and store the master mold in a 4ʺ wafer carrier for future fabrications.C3.3Place the plastic petri dish containing the PDMS mold slab on a cutting mat and set a lamp above the PDMS mold slab that allows the chip perimeter lines to be visible.C3.4Using a scalpel, cut deep into the PDMS mold in straight lines till the scalpel reaches the petri dish surface, allowing clean-cut separation of chips from the PDMS mold slab (obtain 6 chips in total per PDMS mold), as shown in (Figure 5G). Place all 6 chips on another clean plastic petri dish with the chip structures (channels and pillars) facing upwards.C3.5Place the plastic petri dish containing all 6 chips on a cutting mat. For each chip, punch four holes at the locations of media reservoir inlets and outlets using a 4 mm biopsy punch. Similarly, punch six holes at the locations of channel inlets and outlets using a 1 mm biopsy punch. **CRITICAL STEP:** (1) Place a piece of tape on the chip (Figure 5L) to avoid discrepancies (since PDMS is made of transparent plastic with internal reflection of light) in finding the exact location of the holes. (2) Place the biopsy punch vertical to the location of a reservoir inlet. Apply force and press down with the punch (Figure 5M). (3) Lift the chip upwards, as the puncher remains inserted in the chip and press the button on top of the puncher to push out the excess material (Figure 5N). Then, gradually remove the biopsy punch from the chip.C3.6Once punching is complete, gently place a small piece of scotch tape on the chip surface, which will allow any dust or excess PDMS to stick to the tape.C3.7Detach the tape that frees the chip from any physical contaminants. Repeat this step for a total of 3 times for each chip. Place the cleaned chips with the chip structures facing upwards in a glass petri dish sealed with autoclave tape.C3.8Place the sealed petri dish inside a sterilization pouch and dry autoclave at 121 °C for 35 min.

**PAUSE POINT:** These chips can be stored at room temperature for at least one month until plasma treatment in Step C4.

#### 4.3.4. Plasma Bonding of Chips and Cover Glass ● TIMING 10 Min

The purpose of this step is to enclose the chip structure surface with a cover glass and close the system to prevent any reaction of the content of channels with the environment. Plasma cleaning removes organic contaminants and prepares surfaces for subsequent processing through the introduction of chemical functional groups. Plasma cleaning also ionizes the top surface of materials put into the machine. The air inside the plasma cleaner will be ionized to negative ions and will adhere to the top surface of the materials. A pump is attached to the plasma cleaner to create a vacuum so that all the air inside the plasma cleaner can be taken out before treating it with plasma.

C4.1Autoclave the Superfrost glass slides and cover glass in two separate clean pipette tip boxes at 121 °C for 35 min.C4.2Place an autoclaved-sterile Superfrost glass slide on the middle of an empty glass petri dish. Using forceps, place 3 pieces of sterile cover glass on one edge of the Superfrost glass slide as shown in (Figure 5O). Similarly, place 3 chips with the chip structures facing upwards on another edge of the Superfrost glass slide and ensure that there is no contact between the cover glass and the chips (Figure 5O). **NOTE:** Use of a Superfrost glass slide creates contrast between the cover glass and the glass petri dish. In addition, use of a Superfrost glass slide helps prevent attachments of the cover glass or chips with the glass petri dish during plasma treatment.C4.3Place the glass petri dish on the working plate inside the PLASMA CLEANER.C4.4Turn the valve, located on the door of the PLASMA CLEANER, to a closed position and close the door. Turn the PUMP ON and hold the door for at least 5 s since there are no locks for the door and when enough pressure is generated inside the PLASMA CLEANER due to the vacuum, the door will be pressure locked.C4.5Switch the PLASMAFLO ON, also known as pressure monitor. Once the pressure reaches below 400 MTORR, turn the RF level into the MED position in the PLASMA CLEANER and switch the POWER ON for plasma treatment for 2 min. **NOTE:** A violet light will be visible inside the PLASMA CLEANER, which indicates the surfaces of the chips and cover glass are being plasma treated.C4.6After plasma treatment, turn the RF level into OFF position, switch the POWER OFF and then switch the PUMP OFF of the PLASMA CLEANER. Slowly release the vacuum by turning the valve 30–45 degrees counterclockwise. **!CAUTION:** Make sure the pressure release is gradual otherwise the cover glass will be displaced since they are very light and high-pressure difference can induce unwanted movement of cover glass inside the PLASMA CLEANER.C4.7Open the door and carefully remove the glass petri dish and place it on a flat surface.C4.8Acquire an autoclaved-sterile cleanroom wipe and place plasma-treated chips on the white cloth such that the non-plasma treated side is in contact with the wipe.C4.9Use a fine point tweezer to grip one cover glass and place the plasma treated side of the cover glass on the plasma treated side of the chip and then gently press the cover glass using the thumb to enhance attachment. Repeat this process for other chips. **!CAUTION:** Do not press directly on channel areas, otherwise it can rupture the channels. **NOTE:** Make sure there are no bubbles in the interface.C4.10Place the plasma bonded chips in a new cleaned glass petri dish such that the cover glass is in contact with the surface of the glass petri dish.C4.11Repeat step C4.1 to step C4.10 for every 3 chips per plasma treatment.C4.12Place the glass petri dish containing the plasma-bonded PAH-chips in an oven and heat-treat at 65 °C for at least 4 h to further enhance plasma bonding.

**PAUSE POINT:** The plasma bonded chips can be stored at room temperature for at least one month.

### 4.4. Poly-D-Lysine (PDL) Coating and Cleaning of the PAH-Chips

#### 4.4.1. Perform PDL-Coating and Leak Test of PAH-Chips ● TIMING 5 H

The goal of this step is to mix and apply poly-D-lysine (PDL) to coat the chip for enhancing cell attachment. PDL is a synthetic positively charged polymer that enhances electrostatic interaction between negatively charged ions of the cell membrane and positively charged surface ions of attachment factors on the culture surface. When adsorbed to the PDMS surface, PDL increases the number of positively charged sites available for cell attachment. This uniform net positive charge is preferred by collagen and endothelial cell types, which are used in PAH-chips. The entire process of PDL coating is done inside a cell culture hood.

D1.1Using an acetone-cleaned dry spatula, weigh PDL powder at a ratio of 10 mg PDL per 20 microfluidic PAH-chips in a sterile 1.7 mL microcentrifuge tube. **NOTE:** Seal the stock PDL container with parafilm and place it in −20 °C for future use.D1.2Add 1 mL sterile distilled water per 10 mg PDL to prepare 10 mg/mL solution and mix using a vortex mixer for 30 s.D1.3Filter sterilize the PDL solution with a 0.22 μm filter to remove any unwanted or insoluble particles. **NOTE:** If the insoluble particles are not removed, they may clog the chip microchannels, preventing liquid flow and incur leaks.D1.4Place the PAH-chips on a flat surface (a clean pipette tip box can be used) inside a laminar flow hood.D1.5Acquire the 10 μL PDL solution with a sterile P10 micropipette barrier tip that has a tip diameter of roughly 1 mm.D1.6Gently insert the tip into the inlet of the first channel (intimal layer for endothelial cells) such that the tip is circumjacent to the channel inlet and then slowly release the PDL solution till the solution reaches the outlet of the intimal channel. The surface tension due to hexagonal pillars keeps the PDL solution within the intimal layer (as seen with trypan blue solution in Figure 6A,C,E), confirming that the intimal channel has passed the leak test and can be used for seeding PAH endothelial cells in Section E. **!CAUTION**: Carefully lift the tip from the inlet, ensuring the chip does not detach from the cover glass.D1.7Repeat step D1.6 for the third channel (adventitial layer) and check for leakage. Like the intimal layer, the surface tension due to hexagonal pillars keeps the PDL solution within the adventitial layer (as seen with trypan blue solution in Figure 6A,C,E), confirming that this channel has also passed the leak test and can be used for seeding ADCs in Section E.D1.8For the second channel (medial layer for smooth muscle cells), add PDL solution through the channel inlet till the solution travels one-third of the distance of the second channel. The surface films from the first and third channels generate resistance which prevents continuous flow of the PDL solution to the end of the middle channel. Therefore, leave an additional 5 μL solution on the second channel inlet and then place the P10 micropipette tip at the outlet and pull the air through the outlet. This will allow the entire middle channel to be filled with PDL solution. This step confirms the chip has passed the leak test and is suitable for seeding ECs, SMCs, and ADCs in their respective channels. **NOTE:** If any of the channel leaks, the leaked channel cannot be used for cell seeding, because, during seeding cells, one type of cell from a leaked channel will pass to and cross-contaminate the adjacent channel. Indicate the leaked channel with a pen marker. Other channels from the same chip that passed the leak test can be used for cell seeding.D1.9Once the leak test is done, using a sterile P200 micropipette tip, add the PDL solution in the first reservoir inlet and apply pressure by pipetting up and down so that the PDL solution travels across the entire chip including the perivascular and luminal layers.D1.10Repeat step D1.4 to step D1.9 for as many PAH-chips as required.D1.11Place all the PDL-coated PAH-chips inside a humidified chamber (typically a clean pipette tip box can be used filled with sterile water at the bottom) and incubate chips for 4 h at 37 °C or overnight at 4 °C.

#### 4.4.2. Cleaning PDL Solution and Restoring Hydrophobicity of PAH-Chips ● TIMING 24–72 H

After 4 h of incubation, excess PDL solution is cleaned with water which increases hydrophilicity. To keep cells dispersed in the aqueous cell culture medium within the channel, surfaces are made hydrophobic to increase the ‘contact angle’ between the aqueous phase and chip surface. Thus, the chips undergo a drying process (incubation at 85 °C) which restores the hydrophobicity of the chip surface.

D2.1After 4 h or overnight incubation, bring the humidified chamber containing the PAH-chips inside a laminar flow hood.D2.2Hold a chip with one hand and thoroughly aspirate the PDL solution by placing a regular cell culture glass aspirator in channel inlets/outlets and reservoir inlets/outlets.D2.3Using a sterile P200 micropipette barrier tip, immediately add 100 μL sterile deionized water in any reservoir inlet or outlet and apply pressure by pipetting up and down so that the water travels across the entire chip, channel inlets/outlets, and reservoir inlets/outlets. Aspirate water as in step D2.2.D2.4Repeat step D2.3 (washing/aspiration) for two more times. **NOTE:** It is possible that some water remains trapped inside the chips even after aspiration. However, incubation of chips in step D2.6 ensures evaporation of all remaining water.D2.5Repeat steps D2.2 to D2.4 for each PAH-chip.D2.6Place PDL-coated PAH-chips inside a clean pipette tip box and incubate at 85 °C for at least 24 h but no more than 72 h. This will help restore the hydrophobicity of the chips. **NOTE:** Due to PDL treatment, the PAH-chips become hydrophilic, and the cell culture media that is used to seed cells in Section E is also hydrophilic. When two hydrophilic substances meet, the substances tend to spread. Treating the chips with incubation will allow the hydrophilicity of the chip to decrease and increase hydrophobicity to encourage cell adhesion and cell growth on the chips.D2.7After at least 24 h of incubation, place the chips in a new, sterile petri dish and close the lid.D2.8Now place the petri dish under UV light for 1 h and this will ensure sterilization of the chips.D2.9After 1 h, upon observing the chips, if any dust particle remains, use alcohol spray on the chip and clean with tissue without putting any force. The sterile PAH-chips are now ready for cell seeding.

### 4.5. Seeding Cells on the PAH-Chips ● TIMING 3 H

As stated above in the Introduction Section, the PAH-chip is generated to mimic PAH-afflicted pulmonary arteries/arterioles consisting of three layers of cells: endothelial cells (ECs) in the intimal layer, smooth muscle cells (SMCs) in the medial layer, and adventitial cells (ADCs) in the adventitial layer. As demonstrated in Figure 6C–E, pillars between the channels create a surface film that keep liquid in their respective channel. However, seeding cells in all three layers can be challenging, since cells from one layer can pass to adjacent layer, resulting in cross-contamination. To avoid leakage and to generate a proper surface film, first, SMCs (loaded with green Qtracker™ 525) are seeded in the medial layer using a collagen solution that is solidified upon incubation. Afterwards, ECs (loaded with blue CellTracker™ CMHC Dye) and ADCs (loaded with red Qtracker™ 605) are seeded using EGM2 and SmGM cell growth mediums in the intimal and adventitial layers, respectively. Due to gel-solidification of SMCs in the middle channel, ECs or ADCs do not leak or contaminate the medial layer.

E1.1Split cells by following traditional cell splitting protocol for PAH-afflicted cells of pulmonary arteries/arterioles.E1.2Count cells and take 250,000 SMCs, 500,000 ECs, and 250,000 ADCs in three sterile ice-cold microcentrifuge tubes. Centrifuge cells at 533 relative centrifugal force (RCF) for 5 min and keep cells on ice. **NOTE:** In this protocol, these cells were stained with fluorescent cell trackers as per manufacturer instructions.E1.3Preparation of collagen solution: Mix 20 µL of 10× PBS with phenol red, 105 µL of type 1 collagen, 52.5 µL of deionized water, 20 µL of SmGM cell growth medium, and then add 2.5 µL of NaOH to the solution to obtain a 2 mg/mL collagen solution. The phenol red is used as a pH indicator to obtain a pH 7.4 in the collagen solution, which is suitable for cell growth (Figure 6F) **!CAUTION:** Mix in ice and it is easy to create bubbles in the solution, therefore, mix slowly but thoroughly.E1.4For SMCs, remove the supernatant and re-suspend 250,000 SMCs in 50 μL pre-prepared collagen solution (5 × 10^6^ cells/mL). For ECs and ADCs, remove the supernatant and re-suspend 500,000 ECs in 50 μL EGM2 growth medium (no collagen) (10 × 10^6^ cells/mL) and 250,000 ADCs in 50 μL SmGM growth medium (no collagen) (5 × 10^6^ cells/mL). Keep re-suspended cells on ice. **!CAUTION:** Resuspension of SMCs in the collagen solution must be performed on ice so that the collagen solution remains in liquid form. In addition, re-suspension needs to be performed cautiously to avoid formation of bubbles in the collagen solution. **NOTE:** The number of cells listed here are sufficient for seeding cells in 5–6 PAH-chips.E1.5Bring the petri-dish containing UV-treated PAH-chips inside the laminar flow hood and place five PAH-chips on the surface of a clean pipette box.E1.6Using a P10 micropipette, again re-suspend SMCs so that cells are homogenously distributed in the collagen solution (avoid creating bubbles). Draw 10 μL SMC/collagen solution and place the pipette tip in the inlet of the middle channel and slowly push the collagen solution till the solution reaches the outlet of the medial channel. The surface tension due to hexagonal pillars plus the viscosity of the collagen keep SMCs within the medial layer (Figure 7A). **NOTE:** 6–7 μL solution is enough to fill one channel.E1.7Repeat step E1.5 for seeding SMCs in all 5 PAH-chips.E1.8Incubate chips for 20 min. This will allow solidification of the collagen solution.E1.9Repeat steps E1.5 and E1.6 for seeding ECs in the intimal layer. Incubate chips for another 20 min, which will allow the settling down of ECs in the intimal layer (Figure 7B,C).E1.10Repeat steps E1.5 and E1.6 for seeding ADCs in the adventitial layer and incubate chips for another 20 min (Figure 7B,C).E1.11Then, 24 h later, add 100 μL EGM2 medium in both reservoir inlet and outlet of the luminal layer of the PAH-chip (50 μL/reservoir). This will nourish ECs in the intimal layer. Similarly, add 100 μL SmGM medium in both reservoir inlet and outlet of the perivascular layer of the PAH-chip (50 μL/reservoir). This will ensure nourishment of ADCs and SMCs. **NOTE:** The same process (steps E1.5 to E1.10) can be followed for seeding non-PAH cells (control SMCs, ECs, and ADCs) in our PAH-chips for head-to-head comparison of cell growth (diseased versus non-diseased cells), PAH pathophysiology, and testing anti-PAH drugs (please see our recent publication in *Lab on a Chip* [11]).E1.12Place all 5 chips in a cell culture petri dish. In addition, place a cotton soaked with sterile water inside the petri dish to create a humidified chamber.E1.13Incubate the petri dish containing chips at 37 °C with 5% CO_2_ and image every 24 h for up to 5–7 days.E1.14Re-feeding cells every 24 h: Using a P10 micropipette, place the pipette tip in the outlet of the intimal channel and gently pull and discard the old growth medium, then slowly add 6–7 μL fresh EGM2 cell growth medium in the inlet of the intimal channel. Repeat this step for the adventitial layer. Similarly, using a P200 pipette, gently pull and discard old growth medium from all four reservoir inlets and outlets and add 50 μL fresh EGM2 medium/reservoir in both reservoir inlet and outlet of the luminal layer and 50 μL SmGM medium/reservoir in both reservoir inlet and outlet of the perivascular layer.

## 5. Anticipated Results

This manuscript describes a detailed method of design and fabrication of a five-channel PAH-chip device with hexagonal pillars mimicking the perivascular, adventitial, medial, intimal, and luminal layers of the pulmonary artery (Figure 1) [11]. The first major step of this PAH-chip protocol is the preparation of a superior quality master mold, which contains the imprint of the chip design. Here, by following the steps listed in B1 to B6, we have prepared a good quality master mold, as defined by the sharped edges of the pillars and channels and curved inlets/outlets with no inconsistencies (Figure 4H,I).

Once a master mold is prepared, the next major step in the PAH-chip fabrication process is the preparation of the PDMS mold and fabrication of the PAH-chips. It is important to note that one master mold, if its surface remains unaltered, can be used to prepare at least 50 PDMS molds. As reported in our published study [11] and described in steps C1 to C3, we used the photolithography technique to prepare the PAH-chips. After fabricating the chips, we attached them to a cover glass (explained in step C4) and tested them for leakage by passing trypan blue solution through the channels. As illustrated in Figure 6E, there was no leakage of trypan blue solution from the adventitial or intimal channel to the adjacent channel. This indicates the surface tension due to pillars between the channels keeps trypan blue solution within the respective channel, confirming that the PAH-chip passed the leak test and can be used for seeding PAH cells in their respective channels.

We then seeded PAH-afflicted pulmonary arterial endothelial (PAH-ECs), smooth muscle (PAH-SMCs), and adventitial (PAH-ADCs) cells in our PAH-chips (Figure 7) and performed fluorescence microscopy immediately after cell seeding. We observed that PAH-ECs (blue-colored) stayed within the intimal layer and did not leak to the luminal or medial layer (Figure 7A). Similarly, when we seeded PAH-ECs (blue-colored) and PAH-ADCs (red-colored) in the intimal and adventitial layers, respectively, PAH-ECs or PAH-ADCs did not leak to their adjacent channels (Figure 7B). Finally, when we seeded all three cell types such as PAH-ECs (blue-colored), PAH-SMCs (green-colored), and PAH-ADCs (red-colored) in the PAH-chip, they stayed within their respective channels and no cross-contamination of cells was observed (Figure 7C). Unlike the PAH-chip prepared in this protocol, in our published study, the channels were separated by trapezoidal pillars [11], wherein PAH-ECs grew on the intimal layer in a monolayer fashion. Similarly, PAH-SMCs grew in the medial layer and moved to the intimal and adventitial layers. PAH-ADCs also showed growth on the adventitial channel and moved to the neighboring layers [11]. These data show that our PAH-chips support culture of three major PAH-afflicted pulmonary arterial cells. In our published study [11], we also demonstrated that diseased cells seeded in one channel interacted with the cells seeded in the neighboring channels, resulting in an aggressive cell proliferation and aberrant cellular growth, which are characteristic features of pulmonary arterial remodeling [24,25,26]. For instance, in the presence of PAH-SMCs or PAH-ADCs or PAH-SMCs plus PAH-ADCs seeded on their respective channels, PAH-ECs grew aggressively in the intimal channel and moved to the luminal channel, resulting in intimal thickening, a major pathology of the PAH-afflicted pulmonary artery. Importantly, PAH-SMCs (identified by α-SMA expression) migrated from the medial channel into the intimal and luminal channels and interacted with PAH-ECs in the intimal channel. This type of migration/interaction of cells happened in those chips that contain PAH-ECs plus PAH-SMCs or PAH-ECs plus PAH-SMCs plus PAH-ADCs and is more prominent in chips that contain all three types of diseased cells. Migration of PAH-SMCs from one channel to another channel is reflective of arterial thickening or muscularization, a phenomenon that was described previously in PAH-afflicted pulmonary artery of Sugen/Hypoxia-induced animal model of PAH [27]. In contrast, when non-diseased ECs (N-ECs) were seeded in the presence of non-diseased SMCs (N-SMCs) or non-diseased ADCs (N-ADCs) or N-SMCs plus N-ADCs, only a few N-ECs moved from the intimal channel to the neighboring channel and phenotype of N-ECs did not change, as evidence by their lack of α-SMA expression. Similarly, no movement of N-SMCs or N-ADCs were observed from their respective channel to their neighboring channel. Seeding PAH-ECs, PAH-SMCs, and PAH-ADCs in the chips, other major pathologies of PAH and sexual dimorphism were recapitulated [11,12].

Overall, we demonstrated that the microfluidic PAH-chip device can be fabricated in a laboratory setting [11,28,29]. This device allows seeding and culture of three major types of PAH-afflicted pulmonary arterial cells (ECs, SMCs, and ADCs) in their respective channels, and thus mimicking the PAH-afflicted pulmonary artery/arteriole. We believe that the way we described the PAH-chip protocol (design, fabrication, and on-chip cell culture) will help biologists with little or no engineering background to prepare the device for studying PAH pathophysiology and screening anti-PAH drugs in a laboratory setting. We also envision that this protocol can be used to design and fabricate chips to study other vascular diseases.

## Figures and Tables

**Figure 2 micromachines-13-01483-f002:**
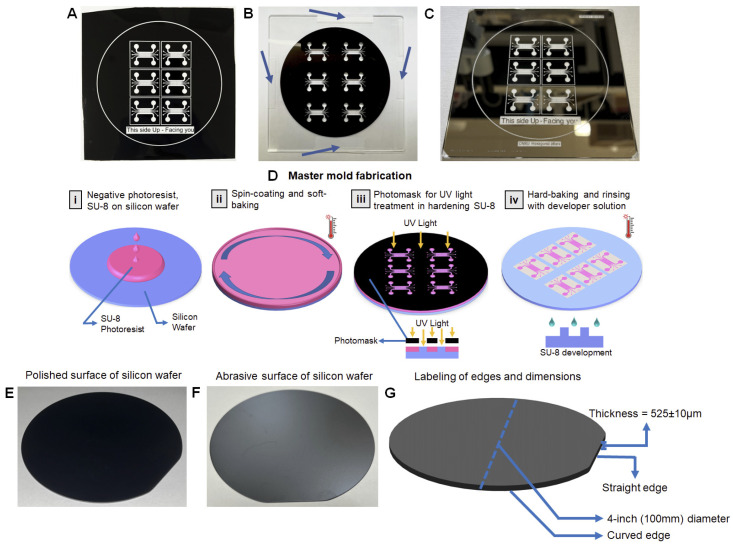
(**A**) Plastic-PAH photomask showing the design of the PAH-chip imprinted on a plastic sheet, which is low in price, supports a low resolution, and has weak stability but it is easy to handle. (**B**) Plastic-PAH photomask attached to thick cover glass that stabilizes the plastic photomask to be repeatedly used during UV-light exposure. (**C**) Chrome-plated PAH photomask. The STL files is directly etched onto the chrome plate surface, which enables very high resolution, and is very easy to clean but more expensive than the plastic photomask. (**D**) A summary of master mold fabrication-(**i**) Pouring SU-8 2100 on the center of the silicon wafer. (**ii**) Use the spin coater to spread the SU-8 evenly across the surface of the silicon wafer and then soft-bake to remove cyclopentanone. (**iii**) UV-light treats the SU-8 surface to imprint and harden the chip design. (**iv**) Hard bake solidifies the UV-treated part of the surface and then use SU-8 developer solution to remove excess untreated SU-8. (**E**) Polished surface of a 4-inch silicon wafer, which gets coated with SU-8. (**F**) Abrasive surface of a silicon wafer: the non-polished back surface of the silicon wafer. (**G**) Labeling of edges and dimensions: reference for common terminologies used in the protocol to describe the features of a silicon wafer.

**Figure 4 micromachines-13-01483-f004:**
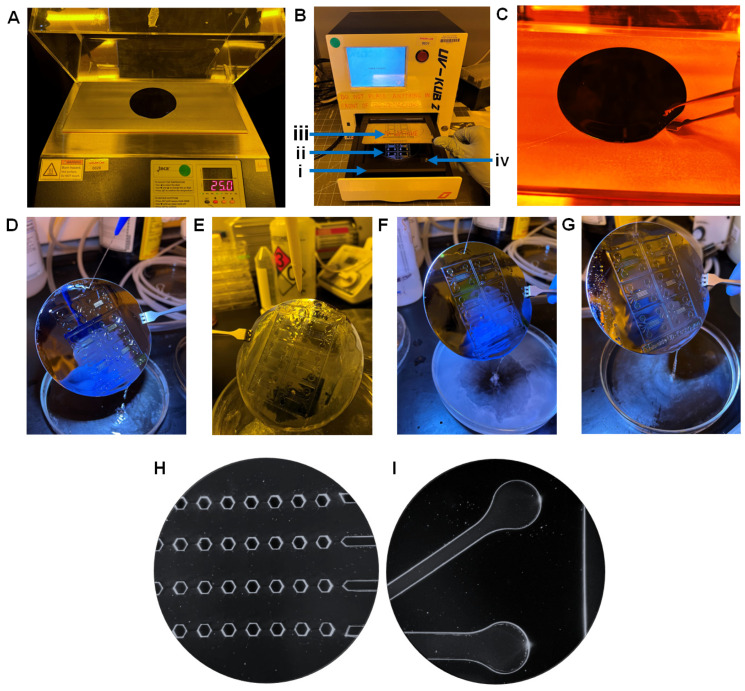
(**A**) Silicon wafer placed on a controllable heat/cool cold plate such that the entire wafer is in contact with the plate. (**B**) Illustration of UV-KUB 2:(**i**) Loading tray—The silicon wafer is placed on the loading tray with the SU-8 covered surface facing up, (**ii**) Soft-baked SU-8 covered silicon wafer—the silicon wafer sits on the loading tray, (**iii**) Chrome Photomask—The chrome photomask is placed on top of the silicon wafer but make sure it does not come into contact with the SU-8 coated side, and (**iv**) Support cylinders—There are two support cylinders diagonal to each other which supports the chrome photomask and moves up and down during calibration before the UV-treatment process begins which controls the resolution of the chip design imprint. (**C**) Handling the silicon wafer during removal—Use the wafer handling tweezers to pry the wafer from all sides in case the wafer is stuck to the cold plate. Then, use the tweezers to grab the silicon wafer from the straight edge only. (**D**–**G**) Washing the hard-baked SU-8 2100 silicon wafer with SU-8 developer solution—(**D**) Hold the silicon wafer slanted with a wafer holding forceps and gently spray developer solution on top of the silicon wafer to allow the developer solution to rinse the untreated, excess SU-8, (**E**) If SU-8 is underdeveloped, white residue starts to form on the silicon wafer when isopropanol is applied, indicating longer development is needed, (**F**) Another sign of underdeveloped SU-8 is the accumulation of white residue in the glass petri dish, and (**G**) Properly developed SU-8 will have no white residue on the silicon wafer, confirming the competition of master mold fabrication. (**H**,**I**) Microscopic views of a silicon wafer—(**H**) The hexagonal pillars have sharp edges indicating high-resolution UV-light treatment and proper SU-8 development, (**I**) The inlets/outlets have curved edges, which indicate no residues or inconsistencies along the edges.

**Figure 6 micromachines-13-01483-f006:**
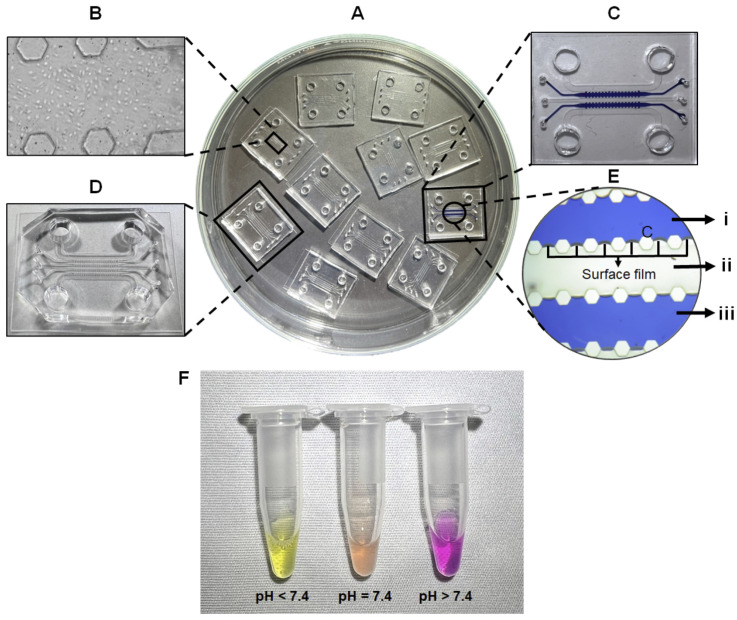
PDMS-based chips—(**A**) 11 chips with five channels and hexagonal pillars. (**B**) Pulmonary arterial endothelial cells seeded on the intimal channel of the chip, the second channel from the top in Panel C. (**C**) Leak test performed by passing a trypan blue solution in the intimal and adventitial channels. The solution did not move from its respective channel to the adjacent channel because of surface-film created by air–water surface tension between pillars. (**D**) Zoomed-in view of a PAH-chip. (**E**) Zoomed-in view of Panel C—(**i**,**iii**) The trypan blue solution in the intimal and adventitial channels in Panel C. (**ii**) The medial channel with no trypan blue solution because the inter-pillar air–water surface film prevented the solution from moving from intimal or adventitial channel to the medial channel. (**F**) pH indicator for collagen solution–—f the pH is less than 7.4, the solution will be yellowish in color and if the pH is more than 7.4, the solution will start to turn pink. The goal is to obtain a light peach color which indicates a pH of 7.4 suitable for cell growth.

**Figure 7 micromachines-13-01483-f007:**
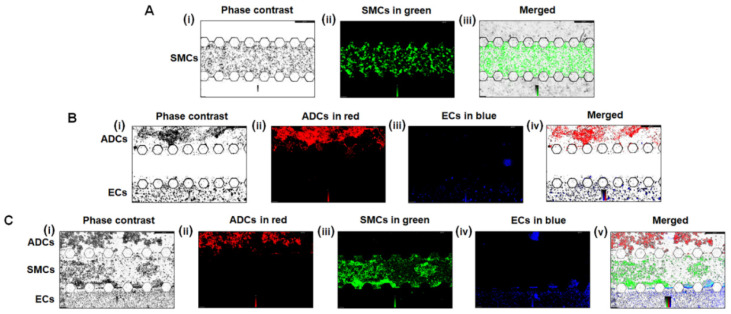
LEICA microscope images of cell seeding in chips with hexagonal pillars. (**A**) Smooth muscle cells (SMCs; loaded with green fluorescence cell trackers) seeded in the medial layer without any leaks into the adventitial or intimal layers, (**i**) Brightfield image of SMCs, (**ii**) Green fluorescence image of SMCs, and (**iii**) Merged image. (**B**) Adventitial cells (ADCs; loaded with red fluorescence cell trackers) seeded in adventitial layer and endothelial cells (ECs; loaded with blue fluorescence cell trackers) seeded in the intimal layer without any leaks in the medial channel or the growth media reservoirs (not shown due to magnification limitations), (**i**) Brightfield image of ADCs and ECs, (**ii**) Fluorescent image of ADCs, (**iii**) Fluorescent image of ECs, and (**iv**) Merged image. (**C**) ADCs seeded in adventitial layer, SMCs seeded in medial layer, and ECs seeded in the intimal layer without any leaks to the growth media reservoirs (not shown due to magnification limitations), (**i**) Brightfield image of ADCs, SMCs and ECs, (**ii**) Fluorescent image of ADCs, (**iii**) Fluorescent image of SMCs, (**iv**) Fluorescent image of ECs, and (**v**) Merged image.

**Table 1 micromachines-13-01483-t001:** Silicon wafers manufactured in different diameters and thicknesses.

Wafer Size	Typical Thickness	Weight Per Wafer	100 MM^2^ (10 MM) Die Per Wafer
1-inch (25 mm)	-	-	-
2-inch (51 mm)	275 μm	-	9
3-inch (76 mm)	375 μm	-	29
**4-inch (100 mm)**	**525 μm**	**10 g**	**56**
4.9-inch (125 mm)	625 μm		95
150-mm (5.9 inch, usually referred to as “6 inch”)	675 μm	-	144
200-mm (7.9 inch, usually referred to as “8 inch”)	725 μm	53 g	269
300-mm (11.8 inch, usually referred to as “12 inch”)	775 μm	125 g	640
450-mm (17.7 inch) (proposed)	925 μm	342 g	1490
675-mm (26.6 in) (theoretical)	unknown	unknown	3427

**Table 2 micromachines-13-01483-t002:** Spin speed, soft-baking conditions, post-exposure baking conditions, and SU-8 development time based on the desired thickness of SU-8 100 or SU-8 2100 on a 4″ silicon wafer.

SU-8	Thickness (µm)	Spin Speed (rpm)	Soft-Baking		Post Exposure Baking (PEB)		SU-8 Development Time (mins)
			Minutes @65 °C	Minutes @95 °C	Minutes @65 °C	Minutes @95 °C	
SU-8 100	100	3000	10	30	1	10	10
	150	2000	20	50	1	12	15
	250	1000	30	90	1	20	20
SU-8 2100	100	3000	5	20–30	5	10–12	10–15
	140	2000	5	20–30	5	10–12	10–15
	265	1000	7	45–60	5	15–20	17–20

**Table 3 micromachines-13-01483-t003:** Time, RPM, and acceleration conditions for each cycle of the spin coater for 150 µm thickness. An acceleration of 2 m/s^2^ is used to indicate no acceleration in the spin coater.

Step	Time (Seconds)	RPM	Acceleration (m/s^2^)
001/005	5	500	100
002/005	10	500	2
003/005	6.6	2000	300
004/005	30	2000	2
005/005	6.6	0	300

**Table 4 micromachines-13-01483-t004:** The ‘set point’, ‘ramp time’, ‘soak time’, ‘function’, and ‘# of repeats’ inputs for soft-baking in Teca software.

Step	Set Point	Ramp Time	Soak Time	Function	# of Repeats
1	65.0	2	30	NEXT	1
2	89.0	2	20	NEXT	1
3	25.0	10	5	END	1

**Table 5 micromachines-13-01483-t005:** The ‘set point’, ‘ramp time’, ‘soak time’, ‘function’, and ‘# of repeats’ inputs for post-exposure baking in Teca software.

Step	Set Point	Ramp Time	Soak Time	Function	# of Repeats
1	65.0	2	5	NEXT	1
2	89.0	2	15	NEXT	1
3	25.0	10	5	END	1

## Data Availability

Not applicable.

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
