# Peer review of "A Protocol for Fabrication and on-Chip Cell Culture to Recreate PAH-Afflicted Pulmonary Artery on a Microfluidic Device"

_micromachines, 2022, doi:10.3390/mi13091483_

Round 1

Reviewer 1 Report

The paper itself is written in good quality. The problem with it is that it is more like a manual, 21 figures and 32 pages are simply too long for a scientific paper. After reading it, I personally think only figures 19, 20, and 21 should be preserved in the paper, the rest of the figures should be combined into figure 1 about the device manufacturing. 

In the current form, I cannot give further reviewer opinion about it and the paper needs significant reform before it can be reviewed as a scientific manuscript. 

Reviewer 2 Report

This manuscript described a detailed method of PAH-chip design and fabrication and cell seeding on the device so that biologists can prepare the device and study PAH pathophysiology. However, some issues need to be solved before receiving.

Major:

1.      The authors mentioned that fluid stress and strain in PAH-on-a-chip help to reproduce physical factors that impact organ functionality, however, this factor is not represented in this manuscript. It is suggested that the author clarify how to simulate the fluid stress and strain in the chip, such as the fluid velocity.

2.      In section D1, the authors used PDL to increase the hydrophilicity of the chip while increasing the number of positively charged sites on the chip. However, in section D2, the authors restored the hydrophobicity of the chip by repeated washing and incubation. Whether the hydrophobic operation will cause the loss of positively charged sites, which will make it difficult for cells to adhere. It is recommended that the authors further clarify how to balance the hydrophilicity and hydrophobicity of the chip.

3.      Due to the existence of liquid fluidity. When seeding cells in each channel, how did the author ensure that the collagen solution (or only cell solution) can fill the whole chamber but not diffuse to the chambers on both sides? What do operators need to pay attention to when seeding cells? For example, is it necessary to control the liquid flow rate? Amount of liquid?

4.      The authors mentioned in the "ANTICIPATED RESULTS" section that the flow channels in the chip were tested with trypan blue staining solution, and the results showed no leakage between the flow channels. In this case, is it possible to directly seed SMC cells without using collagen mixed cells?

5.      Now, there are many protocols about the preparation of PDMS chips. Therefore, it is recommended that the authors clarify how the chip differs in terms of device application and depth of detail in this manuscript.

6.      It is suggested that the authors clarify whether the cells were mixed with collagen when seeding cells in ADC and EC channels.

7.      How did the authors determine the dimensions of the individual channels of the chip? Such as depth, width, etc., these factors have a greater impact on the fluid, so do they have an impact on PAH research?

8.      How did the authors determine the proportions of the three types of cells?

Minor:

1.      EGM2 was repeated twice in the first sentence in E1.11, whether to change one to SmGM medium.

Reviewer 3 Report

In this paper, a PAH chip research model is constructed, and the construction and research methods of the model are described in detail. This study has some reference value for the research in this field. However, some contents in the text need further clarification, such as:

1. What is the design basis of the structural dimensions of channels, pillars and the distance between them, and how to ensure the similarity when simulating PAH?

2. How to evaluate the detection effect of the PAH chip? Is there any comparative analysis of the results of animal experiments?

Round 2

Reviewer 1 Report

The paper is moving in a good direction - however, it still consists of way too many figures and details that can be simply cited from other papers and even test books. The authors need to decide what is novel enough to include in main draft vs. what are things that have been published many times and can be cited without detailing. 

Author Response

There is no question about the novelty of this manuscript, because this is the only paper describing how to prepare PAH-chips mimicking PAH-afflicted pulmonary artery in a lab setting and its possible applications in studying the pathophysiology of PAH. Our goal is for every other PAH investigator or researcher, whoever interested in this device, to make this device in their labs. For this, we strongly believe that removal of any further figures (figure numbers were reduced from 21 to 10 in the first revision, and we have now compromised two images making 9 total figures) or its corresponding details will hamper the reproducibility of making the PAH-chips.

Reviewer 2 Report

This paper mainly described a preparation method of a PAH chip. However, there are many articles describing the preparation methods of PDMS chips, and there is no innovation in the preparation of chips using PDMS materials. The focus of theis article should introduce the innovations in the design and fabrication of this PAH chip, rather than the entire process of preparing the chip from PDMS materials.

Author Response

Our protocol is the only method that describes PAH-on-a-chip device. As such, we have changed our tone in the Introduction section to address novelty– “Although there are several published protocols on polydimethylsiloxane (PDMS)-based chips, no such device is available that recapitulates the PAH-afflicted pulmonary artery. Importantly, our protocol is different from others in terms of application of the device and the depth of the details. For instance, this is the only multichannel chip that represents three major cell layers of a pulmonary artery in addition to luminal and perivascular layers. There are no studies or protocols so far available that provide in-depth explanation of PAH-chip fabrication and on-chip cell seeding.” As per the reviewer’s suggestion, we have replaced the term ‘PDMS chips’ with ‘PAH-chips’ wherever applicable.

Round 3

Reviewer 1 Report

As commented from previous rounds - this paper is rather a manual to producing a microfluidic device that is very generic. Several labs have made almost identical devices, and the manufacturing is clearly well documented (e.g. in [1]). 

Ultimately, I think it is the editor's choice whether this paper fits the journal's scope and format requirements. 

[1] Liu, Yufang, et al. "Angiogenesis and Functional Vessel Formation Induced by Interstitial Flow and Vascular Endothelial Growth Factor Using a Microfluidic Chip." Micromachines 13.2 (2022): 225. 

Author Response

We are aware of this paper. This statement from the reviewer “Several labs have made almost identical devices” is incorrect because there is no microfluidic device available in the field that mimics 5-layers of PAH-afflicted pulmonary artery. More specifically, Yufang et al paper is focused on angiogenesis and has no relation to PAH as a disease and does not represent the 5-layers arrangement of a pulmonary artery (such as luminal, EC, SMC, ADC, and perivascular layers). Thus, we feel that the study performed by Yufang et al is irrelevant in this context.

Reviewer 2 Report

There have been reports about lung airway chips. For example, Sonia Grego et al. introduced the preparation of a bionic airway model in which three primary cells (airway epithelial cells, lung fibroblasts and microvascular endothelial cells) were located in channels separated by three nanopore membranes.  Humayun et al. designed a three-chamber device for simulating a lung airway chip, in which a hydrogel was injected into the middle chamber to act as a barrier between airway epithelial cells and smooth muscle cells. Since this paper focuses on the design and preparation of the lung airway chip, it is better for the authors to clarify the innovations in the design and fabrication of the chip and those reported previously.

Author Response

This statement from the reviewer “Since this paper focuses on the design and preparation of the lung airway chip” is incorrect, because our PAH-chip is not a lung airway chip, rather it is about recapitulating PAH-afflicted pulmonary artery. As we stated in earlier revisions and in the paper itself, we are the first to develop a PAH-chip device to mimic three layers of PAH-afflicted pulmonary artery in addition to luminal and perivascular layers.